# The Harmonic Indel Distance

**Bob Pepin**  *bope@di.ku.dk*
*Department of Computer Science*
*University of Copenhagen*

**Reviewed on OpenReview:** *https: // openreview. net/ forum? id=HV9lXOIZYw*

## Abstract

This short note introduces the harmonic indel distance (HID), a new distance between strings where the cost of an insertion or deletion is inversely proportional to the string length. We present a closed-form formula and show that the HID is a proper distance metric. Then we perform an experimental comparison of HID to normalized and unnormalized versions of the indel distance on benchmark tasks for biomedical sequence data. We finally show planar embeddings of the benchmark datasets to provide some insights into the geometry of the metric spaces associated with the different distance metrics.

## 1 Introduction and Setting

In this paper, we introduce the harmonic indel distance (HID). The HID is a distance metric between strings that is normalized in the sense that two long strings that differ by a single symbol are closer to each other than two short strings that differ by a single symbol. Our main technical contribution is Theorem 3.1 which proves the triangle inequality for HID and shows that it indeed defines a distance metric between strings.

The HID $d(A, B)$ between two strings $A$ and $B$ is defined by

$$d(A, B) = 2H_{|A|+|B|-|\operatorname{lcs}(A,B)|} - H_{|A|} - H_{|B|} \tag{1.1}$$

where $|\cdot|$ denotes string length, $H_n = \sum_{i=1}^{n} 1/i$ is the harmonic series and $\operatorname{lcs}(A, B)$ denotes the longest common subsequence (LCS) of $A$ and $B$. We will also use the notation $\operatorname{scs}(A, B)$ for the shortest common supersequence (SCS) of two strings.

The paper is structured as follows: in the remainder of this section we provide some additional intuition on the definition of the HID, give an overview of related work and briefly comment on the computational complexity. Section 3 proves the triangle inequality. In section 4 we first show that HID can be applied to supervised machine learning tasks by two experiments that apply HID to a classification and a regression task. Next, we show that HID is applicable to unsupervised learning by giving an example of a data visualization task. We additionally compare HID to alternative string distances and show that it differs from alternative distances on some of the tasks. Section 5 wraps up the paper and provides additional suggestions for use cases of the HID.

In order to gain some intuition for the formula (1.1) we can rewrite it as

$$d(A, B) = \sum_{i=|A|}^{|\operatorname{scs}(A,B)|} \frac{1}{i} + \sum_{j=|B|}^{|\operatorname{scs}(A,B)|} \frac{1}{j} \tag{1.2}$$

using that $|\operatorname{scs}(A, B)| = |A| + |B| - |\operatorname{lcs}(A, B)|$. The interpretation is as follows: First we insert characters to transform $A$ into $\operatorname{scs}(A, B)$, where the cost of each insertion is inversely proportional to the length of the intermediate string ($i$ in the formula) at that step. Then we delete characters to transform $\operatorname{scs}(A, B)$ into $B$, with the cost of a deletion again being inversely proportional to the length of the intermediate string ($j$ in the formula) on which it is performed.

Known dissimilarity measures between strings either do not take the length of the strings into account (e.g. Levenshtein distance and its variations) or fail the triangle inequality (e.g. Jaro–Winkler similarity, Sørensen–Dice similarity), see Chapter 11 in Deza & Deza (2013) for an extensive list. In general, the Steinhaus (or *biotope*) transform (Deza & Deza, 2013) can be used to transform an unnormalized metric into a normalized metric while preserving the triangle inequality. We will compare the HID to a normalized distance based on the Steinhaus transform in the experiments in Section 4 and show that it can give different results in some data visualization tasks.

Regarding the computational complexity of computing the HID, observe that the formula (1.1) involves only the harmonic series and the length of the LCS and of the strings $A$ and $B$. The length of a string can be expressed as the length of the LCS of a string with itself. The harmonic series can be precomputed into a lookup table or approximated by the logarithm for large values. The computational complexity of HID is thus equal to the complexity of computing the LCS, which can be computed using dynamic programming algorithms with running time quadratic in the length of the strings (Abboud et al., 2015; Bringmann & Kunnemann, 2015). Recently, algorithms have been developed that approximate the length of the LCS in linear time, which immediately yield linear time approximations of the HID (Bringmann et al., 2023). Other approximations of LCS such as for streaming and spall-space settings (Cheng et al., 2021) or differentiable approximation of the LCS (Yavuz et al., 2018) immediately yield approximations of the HID with the same properties.

## 2 Background and related work

The idea behind edit distances it to define the distance between two strings $A$ and $B$ to be the total cost of transforming $A$ into $B$ through a sequence of operations such as insertions, deletions and substitutions of characters. In this work, we consider so-called indel string distances, which are a particular case of edit distances where the possible operations are restricted to insertions and deletions (indels). In particular, a character substitution corresponds to a deletion followed by an insertion.

The most fundamental indel string distance is simply known as the indel distance (ID), see Deza & Deza (2013). Like the HID, the ID quantifies the total cost of transforming a string $A$ into a string $B$ using the operations of inserting and deleting characters. The cost of each operation for ID is 1, whereas for HID the cost is inversely proportional to the length of the intermediate string. Comparing HID and ID therefore allows to isolate the impact of normalizing the cost. The ID can be computed from the LCS using the formula

$$d_{\text{ID}}(A, B) = |A| + |B| - 2|\operatorname{lcs}(A, B)|.$$

The Steinhaus transform (*biotope transform* in Deza & Deza (2013)) is an alternative way of normalizing string distances. The STID is defined by

$$d_{\text{STID}}(A, B) = \frac{2d_{\text{ID}}(A, B)}{|A| + |B| + d_{\text{ID}}(A, B)}.$$

Denoting $\emptyset$ the empty string, note that $d_{\text{STID}}(A, \emptyset) = 1$ for any string $A$ (since $d_{\text{ID}}(A, \emptyset) = |A|$) and that $d_{\text{STID}}(A, B) \leq 1$ for any $A$, $B$. This shows in particular that $d_{\text{STID}}$ embeds the space of strings of arbitrary length into a sphere of radius 1. The STID is normalized in the sense that if $A$ is a subsequence of $B$ then $d_{\text{ID}}(A, B) = |B| - |A|$ and $d_{\text{STID}}(A, B) = \frac{|B| - |A|}{|B|}$. In contrast for the HID we have $d_{\text{STID}}(A, \emptyset) = H_{|A|}$ so that the space of all strings equipped with the STID distance has infinite radius.

The HID is inspired by the contextualized normalized edit distance from de la Higuera & Mico (2008) which requires the computation of shortest paths over all possible edit operations, implemented using a custom dynamic programming algorithm. The contextualized normalized edit distance is then the sum of the costs over the shortest path, where the cost at each step in the path is normalized by the inverse string length at that step.

From (1.2) it is clear that the HID is identical to the contextualized normalized edit distance restricted to insertions and deletions (there is only a single shortest path). Our closed-form formula allows us to reduce

the computational complexity from cubic to quadratic, thereby answering an open question posed in de la Higuera & Mico (2008). The LCS-based formulation also permits the use of existing libraries for computing LCS, available in all major programming languages, and to leverage efficiency improvements made in the algorithms community as described in the next paragraph. We refer to de la Higuera & Mico (2008) for an overview of the related literature up to 2008. The only subsequent major development known to the authors is the family of extended edit distances developed in Fuad (2014). The basic idea behind the extended edit distances is the addition of a penalty term to an edit distance which needs to be tuned to specific problem instances, in contrast to the parameter-free definition of the HID.

## 3 Proof of main result

**Theorem 3.1.** *The harmonic indel distance*

$$d(A, B) := 2H_{|A|+|B|-|\operatorname{lcs}(A,B)|} - H_{|A|} - H_{|B|}$$

*defines a distance on the space of strings. For any three strings $A, B, C$ it satisfies the distance axioms*

- $d(A, B) = d(B, A)$,
- $d(A, B) = 0 \iff A = B$ *and*
- $d(A, C) \leq d(A, B) + d(B, C)$.

Before proving the theorem, we will start with three preliminary lemmas. In the proofs of the lemmas, we make extensive use of the observation that if $A$ is a subsequence of $B$, then $d(A, B) = H_{|B|} - H_{|A|}$ which follows immediately from the definition.

**Lemma 3.2.** *For any two strings $A$ and $B$*

$$d(A, B) = d(A, \operatorname{scs}(A, B)) + d(\operatorname{scs}(A, B), B).$$

*Proof.* First note that, if $A$ is a subsequence of $B$, then $d(A, B) = H_{|B|} - H_{|A|}$. Now, since $|\operatorname{scs}(A, B)| = |A| + |B| - |\operatorname{lcs}(A, B)|$ we have

$$d(A, \operatorname{scs}(A, B)) + d(\operatorname{scs}(A, B), B) = H_{|\operatorname{scs}(A,B)|} - H_{|A|} + H_{|\operatorname{scs}(A,B)|} - H_{|B|} = d(A, B).$$

$\square$

**Lemma 3.3.** *For any three strings $A, B, C$ such that $A$ is a subsequence of $B$ and $B$ is a subsequence of $C$ we have*

$$d(A, C) = d(A, B) + d(B, C).$$

*Proof.* By the subsequence relations in the assumption the distances simplify and we get

$$d(A, B) + d(B, C) = H_{|B|} - H_{|A|} + H_{|C|} - H_{|B|} = H_{|C|} - H_{|A|} = d(A, C).$$

$\square$

**Lemma 3.4.** *For any two strings $A$ and $B$ we have*

$$d(A, B) \leq d(A, \operatorname{lcs}(A, B)) + d(\operatorname{lcs}(A, B), B).$$

*Proof.* By symmetry, we can assume without loss of generality that $|A| \geq |B|$. Now, by a direct computation

$$
\begin{aligned}
&d(A, B) - d(A, \operatorname{lcs}(A, B)) - d(\operatorname{lcs}(A, B), B) \\
&= 2H_{|A|+|B|-|\operatorname{lcs}(A,B)|} - H_{|A|} - H_{|B|} - (H_{|A|} - H_{|\operatorname{lcs}(A,B)|}) - (H_{|B|} - H_{|\operatorname{lcs}(A,B)|}) \\
&= 2(H_{|A|+|B|-|\operatorname{lcs}(A,B)|} - H_{|A|}) - 2(H_{|B|} - H_{|\operatorname{lcs}(A,B)|}).
\end{aligned}
$$

The first expression in parentheses is a sum of $|B| - |\operatorname{lcs}(A,B)|$ terms, all of which are less or equal than $1/|A|$ so that

$$H_{|A|+|B|-|\operatorname{lcs}(A,B)|} - H_{|A|} = \sum_{i=|A|}^{|A|+|B|-|\operatorname{lcs}(A,B)|} 1/i \leq \frac{|B| - |\operatorname{lcs}(A,B)|}{|A|}.$$

The second expression in parentheses above is also a sum of $|B| - |\operatorname{lcs}(A,B)|$ terms, all of which are greater or equal than $1/|B|$, and by our assumption $1/|B| \geq 1/|A|$. Therefore

$$H_{|B|} - H_{|\operatorname{lcs}(A,B)|} = \sum_{i=|\operatorname{lcs}(A,B)|}^{|B|} 1/i \geq \frac{|B| - |\operatorname{lcs}(A,B)|}{|B|} \geq \frac{|B| - |\operatorname{lcs}(A,B)|}{|A|}.$$

This shows that

$$d(A,B) - d(A,\operatorname{lcs}(A,B)) - d(\operatorname{lcs}(A,B),B) \leq 0$$

which is the conclusion. $\square$

*Proof of Theorem 3.1.* The symmetry $d(A,B) = d(B,A)$ follows trivially from the definition. So does the "if" direction of the second property. For the only if direction, note that $d(A,B)$ can be decomposed into a sum of two positive terms

$$d(A,B) = (H_{|\operatorname{scs}(A,B)|} - H_{|A|}) + (H_{|\operatorname{scs}(A,B)|} - H_{|B|})$$

so that $d(A,B) = 0$ implies $|\operatorname{scs}(A,B)| = |A| = |B|$ and $A = B$. To prove the triangle inequality, we use the notation and property $\operatorname{scs}(A,B,C) = \operatorname{scs}(A,\operatorname{scs}(B,C)) = \operatorname{scs}(\operatorname{scs}(A,B),C)$ and apply in turn Lemmas 3.2, 3.4, 3.3 twice and 3.2 again to obtain

$$
\begin{aligned}
d(A,B) &+ d(B,C) \\
&= d(A,\operatorname{scs}(A,B)) + d(\operatorname{scs}(A,B),B) + d(B,\operatorname{scs}(B,C)) + d(\operatorname{scs}(B,C),C) \\
&\geq d(A,\operatorname{scs}(A,B)) + d(\operatorname{scs}(A,B),\operatorname{scs}(A,B,C)) + d(\operatorname{scs}(A,B,C),\operatorname{scs}(B,C)) + d(\operatorname{scs}(B,C),C) \\
&= d(A,\operatorname{scs}(A,B,C)) + d(\operatorname{scs}(A,B,C),C) \\
&\geq d(A,\operatorname{scs}(A,C)) + d(\operatorname{scs}(A,C),C) \\
&= d(A,C).
\end{aligned}
$$

$\square$

## 4 Experiments

The purpose of the experiments in this section is to compare the HID to other string distances when applied to machine learning tasks: the indel distance (ID) and the Steinhaus transform indel distance (STID). Note that we do not compare against the contextualized normalized edit distance since it is identical to the HID if we restrict the operations to insertions and deletions as we do in this paper. In addition, the cubic complexity of the normalized edit distance would be prohibitive for our experiments.

We perform experiments on two benchmark tasks for biological sequence regression and classification respectively. We also present planar embeddings for each dataset using t-SNE to gain some insight into the geometries of the spaces associated to the different distances. Table 1 presents statistics on the datasets used. As our main goal is to evaluate the differences between HID, ID and STID, we do not aim to beat state-of-the-art deep learning models but we do include standard baselines to put our results into perspective. We do not claim either that HID is inherently superior to any of the other distances. Each distance embeds the data in a metric space with a different geometry, and the most appropriate geometry depends on the task at hand. Our experiments do show that the different metrics considered result in differences in performance for some but not all tasks and that normalization by string length is beneficial for the two supervised learning tasks considered.

All benchmark experiments used support vector machines with radial basis function kernels based on the string metrics described above. The SVM margin as well as the RBF variance hyperparameters were optimized using the Tree-structured Parzen Estimator algorithm implemented in the Optuna software (Akiba et al., 2019). We used the SVM implementation from Scikit-Learn (Pedregosa et al., 2011). The hyperparameters used are included in Table 4 in the appendix.

Table 1: Dataset statistics

| Dataset | Number of Sequences | | | Sequence Length | | |
|---|---|---|---|---|---|---|
| | Training | Validation | Test | Min. | Max. | Median |
| ncRNA | 25371 | 6430 | 13646 | 42 | 500 | 123 |
| FLIP Mixed | 22335 | 2482 | 3134 | 20 | 35213 | 413 |
| FLIP Human | 7287 | 861 | 1945 | 39 | 34350 | 477.0 |
| FLIP Human-Cell | 5149 | 643 | 1366 | 44 | 34350 | 469.0 |

### 4.1 Classification

The classification task involves the classification of sequences of non-coding RNA according to their type and uses the *Dataset2* dataset from the benchmark paper Creux et al. (2024). This dataset was chosen because non-coding RNA are some of the shortest biological sequences, and we expect the benefit of the normalization in HID to be higher for shorter sequences. The dataset provides training and test splits, and a validation set was generated by random splitting of the provided training set.

The results are shown in Table 2. We see that the SVMs with HID and STID kernels are competitive with the strongest baseline, which uses a recurrent neural network architecture, whereas the SVM with ID kernel underperforms 4 out of the 6 baselines whereas .

Table 2: Classification accuracy on ncRNA benchmark

| Model | Accuracy (%) |
|---|---|
| SVM (HID) | **97.3** |
| SVM (STID) | 97.0 |
| SVM (ID) | 90.6 |
| ncrna-deep | 97.1 |
| MFPred | 96.5 |
| RNAGCN | 94.7 |
| NCYPred | 91.6 |
| nRC | 78.3 |
| ncRDense | 73.5 |

Baseline values are taken from Creux et al. (2024)

### 4.2 Regression

We evaluate the regression performance on the thermostability prediction task from the FLIP benchmark for protein sequences (Dallago et al., 2021). This is a challenging benchmark which includes a carefully selected train-validation-test split based on biological considerations. The metric adopted by the benchmark is the Spearman correlation coefficient. The baselines include language models pre-trained on a large corpus of sequence data (ESM), the same models trained only on the benchmark training set (ESM-untrained) as well as a CNN and a ridge regression model.

Our results are presented together with the baselines in Table 3. The SVMs using normalized distances HID and STID are competitive with all baselines that have not been pre-trained on a large corpus of external sequence data, whereas the unnormalized ID SVM only outperforms the weakest baseline.

Table 3: Spearman correlation coefficient on FLIP thermostability prediction task

| Model | Mixed | Human | Human-Cell |
|---|---|---|---|
| *Models pretrained on large corpus* | | | |
| ESM-1b (per AA) | **0.68** | 0.71 | 0.76 |
| ESM-1v (per AA) | 0.65 | **0.77** | **0.78** |
| *Models without pretraining* | | | |
| SVM (HID) | 0.42 | **0.59** | 0.56 |
| SVM (STID) | 0.40 | **0.59** | **0.57** |
| SVM (ID) | 0.20 | 0.39 | 0.40 |
| ESM-untrained (per AA) | **0.44** | 0.44 | 0.46 |
| ESM-untrained (mean) | 0.36 | 0.48 | 0.49 |
| CNN | 0.34 | 0.50 | 0.49 |
| Ridge | 0.17 | 0.15 | 0.24 |

Baseline values are taken from Dallago et al. (2021)

## 4.3 Metric Embedding

To give some more insights into the different geometries entailed by the HID, STID and ID, we provide t-SNE plots of the training datasets for ncRNA (Figure 1) and FLIP (Figure 2). The objective of t-SNE is to embed a dataset into the plane while keeping the distances in the embedding as consistent as possible with the distances in the original space. Each of the distance metrics defines a metric space of strings, and we expect the t-SNE embedding to reflect as much as possible the geometry of the dataset, viewed as points in the string space entailed by the respective distance metric.

For the ncRNA dataset, a visual inspection of Figure 1 suggests that all distances result in a similar geometry, with the normalized distances leading to a slightly sharper separation for example between red and green classes.

On the other hand, for the FLIP datasets (Figure 2) the different distances lead to clearly different embeddings. The HID leads to a dataset geometry that can be embedded as a crescent shape, with lower-valued points concentrating in one end and higher-valued points concentrating in the other end, especially for the Human and Human-Cell datasets.

The embedding for STID still shows concentration of high- and low-value points in different regions of the embedding space and recovers the same local structures as the HID embedding. However, it does not show any non-trivial global structures, which might be due to the geometry of the STID metric space being less compatible with Euclidean plane geometry than the HID metric space (recall that the STID space is a sphere of radius 1 where all elements are at distance at most 1 from each other). This interpretation is supported by the STID embedding having significantly higher KL divergence than the HID embedding in the t-SNE objective (Table 5 in the appendix).

Finally, the ID produces a radially symmetric embedding with regularly spaced patterns that do not show any obvious relation to the target value. The regular spacing could be caused by the large number of points that each are at the same integer distance from each other.

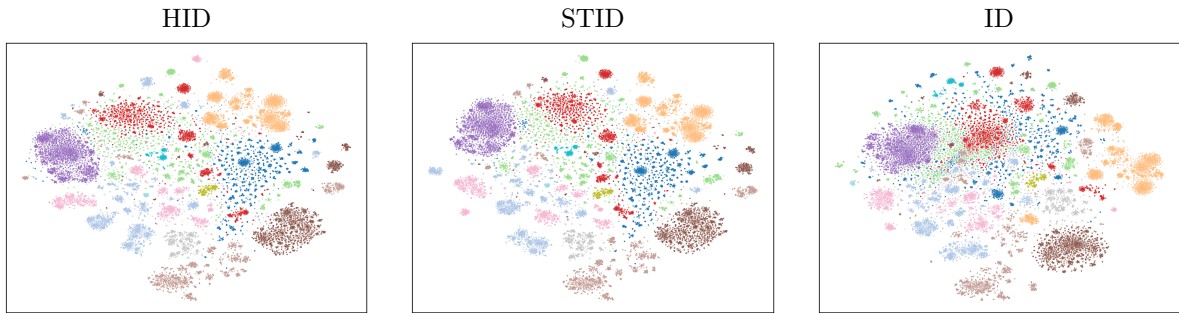

Figure 1: T-SNE plots of ncRNA training dataset using different distance metrics. Colors correspond to different classes. All metrics recover the clusters present in the data with HID and STID obtaining a slightly better separation.

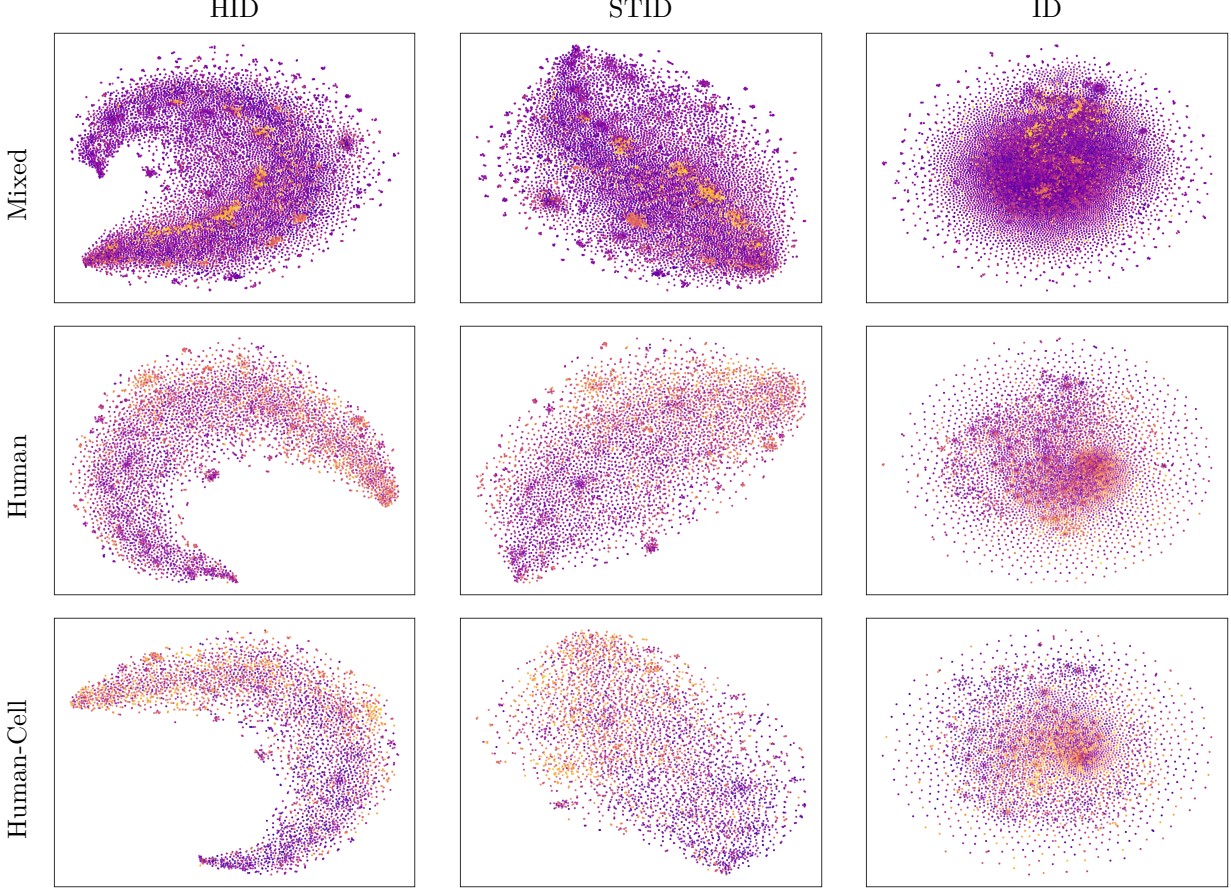

Figure 2: T-SNE plots of FLIP training datasets using different distance metrics. Colors correspond to different target values for thermostability in the regression task. HID shows both global and local structure, STID shows local structure and ID shows little apparent structure.

# 5    Discussion

We introduced the harmonic indel distance and showed that it defines a distance metric. We showed that the harmonic indel distance outperforms the unnormalized indel distance on two biomedical sequence regression and classification tasks while showing comparable performance to a normalized version of the indel distance. Our experiments on planar embeddings with t-SNE show that HID and STID can in some, but not all, cases result in different planar embeddings. The original motivation for the development of the HID was classification of web browsing histories, which involves significantly shorter data with more variation in the length of sequences, where the lack of normalization in ID is expected to have an even larger impact. It is striking that there are still such large differences in performance on datasets where the median length is in the hundreds of characters, and it could be interesting to evaluate the HID on shorter sequences, for example within social sequence classification. Unfortunately, as of the time of submitting this manuscript, we were not aware of any suitable benchmark datasets with baselines containing short sequences with high variability in sequence length.

### Acknowledgments

The author acknowledges funding received under European Union's Horizon Europe Research and Innovation programme under grant agreement No. 101070408.

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

## A  Experimental Details

The SVMs used in the experiments have regularization parameter $C$ and the kernels used are $k(A, B) = e^{-\gamma \operatorname{dist}(A,B)^2}$ where dist is either HID or ID and with the values for $C$ and $\gamma$ given in Table 4.

All t-SNE embeddings used a target perplexity of 30 and were run for the number of iterations given in Table 5.

Table 4: Hyperparameters

| Dataset | Distance Metric | $\gamma$ | $C$ |
|---|---|---|---|
| ncRNA | HID | 8.443116755229333 | 99.5612463103948 |
| ncRNA | STID | 9.91793899262693 | 1.485844877379199 |
| ncRNA | ID | 0.00012069602683643651 | 0.3910622178775264 |
| FLIP Mixed | HID | 2.694680159717171 | 5.303192435782873 |
| FLIP Mixed | STID | 3.4503356492866195 | 1.696222220623828 |
| FLIP Mixed | ID | 0.00014856111427324835 | 2.018895367718889 |
| FLIP Human | HID | 3.0711811333241985 | 8.972094031636633 |
| FLIP Human | STID | 2.2338162722360013 | 2.472937362472116 |
| FLIP Human | ID | 0.0001036128449343376 | 0.6548096783807119 |
| FLIP Human-Cell | HID | 2.211198503980429 | 5.994311048805454 |
| FLIP Human-Cell | STID | 4.758786457584244 | 29.469372386529603 |
| FLIP Human-Cell | ID | 0.00010519254700762129 | 0.02074221663631553 |

Table 5: Hyperparameters and statistics for t-SNE plots

| Dataset | Distance Metric | Iterations | KL Divergence |
|---|---|---|---|
| ncRNA | HID | 2048 | 1.876777 |
| ncRNA | STID | 2048 | 1.889551 |
| ncRNA | ID | 2048 | 2.277497 |
| FLIP Mixed | HID | 8192 | 2.349853 |
| FLIP Mixed | STID | 8192 | 2.582142 |
| FLIP Mixed | ID | 8192 | 3.670377 |
| FLIP Human | HID | 8192 | 1.968790 |
| FLIP Human | STID | 8192 | 2.299917 |
| FLIP Human | ID | 8192 | 3.546993 |
| FLIP Human-Cell | HID | 8192 | 1.891862 |
| FLIP Human-Cell | STID | 8192 | 2.250279 |
| FLIP Human-Cell | ID | 8192 | 3.459189 |

### A.1    Effect of perplexity parameter on t-SNE plots

To validate that the t-SNE plots in the main paper are representative, we here show the plots resulting from a parameter sweep of the perplexity parameter from $2^{-2}$ to $2^{1}2$ for the Human Cell dataset from FLIP. The plots show that our observation that HID preserves more global structure in the t-SNE plots than STID is robust to different parameter values. All embeddings were run to convergence.

The resulting plots are included as a separate file in the supplementary material.

