# OpenReview forum: "The Harmonic Indel Distance"
_TMLR — Accepted by TMLR_

### Review · Reviewer_5NUi · 2024-08-29

**Summary Of Contributions:**

This paper presents a novel normalized string indel distance metric called the Harmonic Indel Distance (HID). In HID, the cost of insertion or deletion is inversely proportional to the string length. The authors demonstrate that HID meets the required distance axioms. They then compare HID with the normalized metric STID and the unnormalized metric ID using biomedical datasets. Lastly, the authors illustrate the planar embeddings of the benchmark datasets to explore the geometric properties of HID, STID, and ID.

**Audience:**

Yes

**Claims And Evidence:**

Yes

**Requested Changes:**

1. Regarding the special contribution of HID, the performance of HID seems to be the same as STID according to the recorded experiment results.

2. The authors mention that they do not aim to beat SOTA method but to evaluate the differences between HID, STID, and ID. However, the authors only investigate the difference in t-SNE embedding shape. According to prior work[1], the global structure of the t-SNE plot may vary according to different parameter settings. Therefore I would suggest a deeper investigation into this question.

[1] Wattenberg, Martin; Viégas, Fernanda; Johnson, Ian. How to Use t-SNE Effectively. https://distill.pub/2016/misread-tsne/

**Strengths And Weaknesses:**

The proposed HID method is a contribution to indel distance metrics. HID satisfies the distance axioms and has comparable performance with mainstream methods like STID.

---

> ### Author Response · Authors · 2024-09-25
> **Individual Response**
>
> **Changes**
>
> 1) No performance advantage
>
> No advantage claimed, see common response.
>
> 2) T-SNE hyperparameter investigation
>
> Added, see common response.

---

### Review · Reviewer_kjbY · 2024-09-05

**Summary Of Contributions:**

The paper introduces the Harmonic indel distance (HID), which is a normalized version of the edit/indel distance. HID builds on the harmonic series and has low compuational complexity, reducible to calculating the longest common substring between the the two input strings.

The paper theoretically proves that HID is a distance metric and additionally provides empirical experiments for biological sequence regression and classification, as well as an analysis for trained embeddings using t-SNE. The used datasets are "Dataset2" from a non-coding RNA classification benchmark (ncRNA) and the FLIP benchmark. The machine learning model is a SVM with radial basis function kernels.  HID is compared to the Steinhaus indel distance and indel distance, as well as established competitors of other approaches for the respective tasks.

The results show that HID for SVM training is superior or at least competitive in the classification and regression tasks. It specifically outperfors the indel distance for classification, while being roughly similar in performance to the Steinhaus variant. The resulting embeddings show to also preserve the global structure of the dataset better than the competitors in the given experiment.

**Audience:**

Yes

**Broader Impact Concerns:**

Broader impact concerns are not available in the paper, but it is deemed ok that way. The paper proposes a novel distance metrics suitable for string representations, which can be applied broadly. The proposed distance metrics does not introduce any obvious biases which might have general manipulative/negative repercussions.

**Claims And Evidence:**

Yes

**Requested Changes:**

* Please provide a dedicated related work or background section with edit distance, contextualized normalized edit distance, Steinhaus transform indel distance, ... . While the introduction has a short paragraph on related works, it does not clarify differences to direct competitors.
* Please provide a more detailed discussion why a single perplexity parameter is sufficient to claim that global structure can be better preserved using HID.
* Please provide a reference for the used and mentioned Optuna software as well as the Scikit-learn library.
* Please provide a motivation why an empirical evaluation against the contextualized normalized edit distance is not needed or add it to the paper. The paper already mentiones downsides of the work (i.e., needing to calculate shortest paths), but it cannot be ruled out that there might be predictive performance differences for downstream tasks.

**Strengths And Weaknesses:**

# Strenghts
- Sound paper, providing theoretical proofs for the distance metric as well as insightful empirical experiments
- Sufficiently different to related works, extending prior works on normalized edit distances with a novel grounding based on the harmonic series.

# Weaknesses
- The embeddings experiments could benefit from varying the used perplexity of 30 or at least better discussing why it is sufficient to not do so. The question remains open if the global structure might be better preserved for the other metrics when using another perplexity parameter.
- Empirical added value compared to existing normalized edit distances is rather low
- Not clear from the paper why the empirical evaluation does not include the referenced work of contextualized normalized edit distance.

---

> ### Author Response · Authors · 2024-09-25
> **Individual Response**
>
> **Weakness:**
> 1) Dependence on Perplexity
>
> Added extra experiments. See common response.
>
> 2) Not significant for applications
>
> We do not claim it is significant. See also common response.
>
> 3) Why not included contextualized normalized edit distance
>
> Added paragraph explaining that. See common response.
>
> **Changes:**
> 1) Dedicated background section
>
> Added further clarification on difference to contextualized normalized edit distance.
> Merged the discussion of ID and STID from the Experiments section with the related works paragraph into a new dedicated background section
>
> 2) Perplexity
>
> See common response, added parameter sweep
>
> 3) References for Sklearn and Optuna
>
> Added
>
> 4) Motivation for why not evaluation against contextualized normalized edit distance
>
> Added, see common response

---

### Review · Reviewer_SHHu · 2024-09-23

**Summary Of Contributions:**

In this paper the authors describe a new similarity between two strings, called the harmonic indel distance. Their metric is similar to edit distance but penalizes edits less for larger strings. The authors prove their similarity is a metric and show some usecases for it on classifying ncRNAs and thermostability prediction for proteins.

**Audience:**

Yes

**Claims And Evidence:**

Yes

**Requested Changes:**

I would like to see a modified manuscript that address the major weaknesses I described above. In particular to secure my recommendation for acceptance, I would like to see: (1) more evaluation on real data showing how/why the HID improves over the STID/ID; (2) the metric embedding section should be removed; and (3) more discussion on theoretical properties of HID, in particular describing how to compute the weighted HID.

**Strengths And Weaknesses:**

Strengths:

- Authors derive novel metric between strings

Weaknesses:

- Authors do not motivate why a different string metric is needed and empirical evaluation is not convincing
    - The empirical results show marginal benefit over other metric (STID) and other existing methods
    - The paper would benefit from more exploration into the specific settings in which HID has improved performance over STID. e.g. in the empirical evals, the authors could show whether HID does have higher accuracy for shorter strings, as suggested by the Intro, or for ncRNAs/proteins of a specific type?
- Metric embedding section is not rigorous and does not provide any insight
    - I do not think it is that useful to look at t-sne embeddings. These embeddings are known to be sensitive to hyperparameters and not properly reflect distances (e.g. Chari and Pachter, Plos Comp Bio 2023, among many other papers). These embeddings in Figs 1/2 just show that the HID is different, and do not provide any insight into why the HID is different (or better) than the STID or ID metrics
- More theoretical characterization is necessary
    - The authors make a small comment about the HID corresponding to a restricted version of the normalized edit distance from de la Higuera & Mico (2008). This should be described in greater detail.
        - It would also be useful to describe the *weighted* HID, where certain edits are penalized more than others. Such weighted distances are often used to quantify the distance between DNA motifs (eg Gupta et al, Genome Biol 2007)
    - Moreover, the authors could give numerical examples (e.g. on simulated data) showing how the HID and STID differ for short strings versus larger strings
    - The HID also seems non-ideal for very long strings, eg genome-length strings, since one would presumably need many bits to store 1/i for very large i. It would be useful to see some discussion on how to represent the HID for long strings.

---

> ### Author Response · Authors · 2024-09-25
> **Individual Response**
>
> Thank you for the detailed review. We have tried to rephrase your comments to make sure we correctly understood them and address them one at a time below. Please let us know if you think we misunderstood some of your comments.
>
> **Weaknesses:**
>
> 1a) It is not clear why HID is significant for applications.
>
> Answer: We do not claim significance. See common response. Please let us know if you still find a passage in the manuscript that might be misinterpreted as claiming significance to a particular application domain.
>
> 1b) Empirical evaluation not convincing.
>
> The purpose of the empirical evaluation is to show that there exist machine learning tasks to which HID is applicable. We acknowledge that some parts of the original manuscript could have been misleading in that sense (especially the Discussion section), and we hope that the revised manuscript is clearer.
>
> 2a) T-SNE embeddings are sensitive to hyperparameters
>
> See common response.
>
> 2b) T-SNE embeddings do not properly reflect distances.
>
> The t-SNE objective explicitly optimizes for matching distances in the original space and the plane. Distortion of distances is of course inevitable, as the paper you reference also points out. The perplexity hyperparameter controls this inevitable trade-off between preserving local and global geometry. The purpose of our t-SNE plots is to provide evidence that the difference in geometry induced by the different norms is not merely a mathematical curiosity but can also have a downstream effects in applications.
>
> As far as we can tell, the paper you reference reports the results from experiments embedding some biomedical datasets and is mostly concerned about whether such embeddings should be used for decision making in the biomedical domain. We do not advocate for any particular decision making process in any particular application domain.
>
> 2c) The embeddings only show that HID is different.
>
> This is exactly our claim. We have added some extra language to clarify this particular claim as explained in the common response. In addition we formulate a hypothesis on why this might be the case in subsection 4.3. We added a comment on the difference between the geometries induces by HID and STID in the new section on Background and related work. In the concrete case of Fig 2, we also point out in the legend to that figure how the HID differs from STID and ID in that it preserves more of the geometry, global and local vs. local only.
> We acknowledge however that the use of the word “recovers” in the legend of Figure 2 might have been suggesting that the same global structure is present in both metric spaces and the differences are due to the difference in distortion caused  by the embeddings. We have replaced “recovers” with “shows” to avoid making any definite conclusions on the metric space.
>
> 3a) More detailed comparison to normalized edit distance
>
> Addressed in revised manuscript
>
> 3b) Possible extension to weighted edit distance
>
> Thank you for the suggestion. It might be possible to weight insertions and deletions differently by adding weights in front of the sums in equation (1.2). However, it is not clear if the triangle inequality would still hold. Showing this would require a reworking of the proof, requiring a major revision of the paper. As such, we consider it out of scope of the present work, but it could be interesting future work if the HID turns out to be useful in application domains that use weighted distances.
>
> 3c) Short strings vs. longer strings on simulated data
>
> Thank you for the suggestion. We feel that the claims of the paper paper (triangle inequality, existence of machine learning applications, different from STID in some cases) are sufficiently supported by the existing evidence. If you disagree, could you please point out which concrete claim would require the experiments on simulated data?
>
> 3d) Applications to long strings (e.g. genomes)
>
> As described in the paragraph on computational complexity, the harmonic series can be approximated by the logarithm for large values. In this case, one would also use one of the near-linear time approximations as described in the same paragraph.
>
> **Requested Changes:**
>
> 1) Show how/why the HID improves over the STID
>
> We do not claim that it improves over the STID. We made some changes to the manuscript to clarify out claims. Please let us know if you think some sections of the new manuscript still suggest such a claim.
>
> 2) Remove metrical embedding section
>
> This section is essential for our claim that the HID is different from the STID in some applications and also in order to support our discussion of the different geometries entailed by the different metrics, as well as showing that this difference can have downstream effects.
>
> 3) More discussion on theoretical properties, weighted HID
>
> Weighted HID is unfortunately out of scope, as discussed above. Could you elaborate on which theoretical properties you think are missing to support the claims of the paper?

---

### Author Response · Authors · 2024-09-25
**Common Response**

We thank the reviewers for their careful reading of the manuscript and their suggestions for improvements. Certain points have been made by all the reviewers, which we address in this common response.

Before doing so, we would like to reiterate the claims we make in the paper.

Our main claim is:
* **Claim 1:** The HID satisfies the triangle inequality. We support this claim by a mathematical proof.

We think this result is both novel and non-trivial.

Our secondary claims are:
* **Claim 2:** There exist machine learning tasks to which the HID is applicable. We support this claim by providing an example of a supervised and an unsupervised task.
* **Claim 3:** There exist machine learning tasks where the HID differs from the STID. We support this claim by providing an example of such a task in the unsupervised learning section on t-SNE.

The associated experiments are essentially routine exercises.

We now address the points made by the majority of reviewers and describe how we addressed them in the updated manuscript.

**Point 1:** All of the reviewers noted that it is not clear that the HID leads to improved performance in applications compared to existing distance metrics.

**Answer:** We do not claim that the HID leads to performance improvements.

**Action:** We have updated the outline of the paper in the introduction stating more clearly the claims made and added a paragraph at the beginning of the experimental section emphasizing which claims we do make and which claims we do not make. We also rewrote the discussion section, which might have been misleading the reader into thinking we made broader claims than we intended to make.

**Point 2:** It is not clear whether the differences in the t-SNE plots are due to the choice of hyperparameter.

**Answer:** In principle, the two relevant hyperparameters are the number of iterations and perplexity. Since we let all of the t-SNE embeddings iterate to convergence, the only remaining hyperparameter is perplexity.

**Action:** We added a section to the appendix with a sweep of different perplexity values for one of the FLIP datasets. Since the results for all three FLIP datasets were qualitatively similar and the presented dataset is a subset of the other datasets, we expect that this is representative for all three datasets. We perform the sweep in powers of two, which corresponds to a linear increase in entropy. Since the embeddings do not change anymore for at least two sequential perplexity values in both extremes, we estimate that we covered the whole range of relevant values.

**Point 3:** Why are we not comparing against contextualized normalized edit distance? Please provide more background on related distance metrics.

**Answer:** In fact, the HID is identical to the contextualized normalized edit distance when restricted to insert and delete operations. Additionally, the contextualized normalized edit distance has cubic complexity, making it prohibitively expensive for most practical applications. This was also noted by the paper originally introducing it and the authors raised an open question on whether there exists approximations or variants with quadratic complexity. We answer that open question in this work.

**Action:** To better explain this point, we added a new section “Background and related work” under “Introduction” that contains a short introduction to edit distances. We also moved the description of ID and STID from the Experiments section to this new subsection. Finally, we elaborate on the difference with contextualized normalized edit distance and highlight the point on the quadratic vs. cubic computational complexity. We also added a remark at the beginning of the Experiments section reiterating the reasons for not including contextualized normalized edit distance in the comparison.

---

### Decision · Action_Editor_E6Fp · 2024-11-21

**Recommendation:** Accept with minor revision

**Comment:**

All of the reviewers agree the correctness and originality of the paper. The applications provided in the paper are also great. Please update the manuscript based on the feedback from the reviewers during the discussion phase for the camera ready.

**Audience:**

The HID is readily applicable to common ML tasks including classification, regression, and metric embedding. The topics are relevant to TMLR.

**Claims And Evidence:**

The paper proposes the harmonic indel distance and formally shows that it satisfies the metric axioms and triangle inequality. The effectiveness of the HID is supported by the experiments with biomedical datasets.